

# From "weak" to "strong" hole confinement in a Mott insulator

Krzysztof Bieniasz[1,2], Piotr Wrzosek[3], Andrzej M. Oleś[2,4], Krzysztof Wohlfeld [3*]

**1** Stewart Blusson Quantum Matter Institute, University of British Columbia,
Vancouver, British Columbia, Canada V6T 1Z4
**2** Marian Smoluchowski Institute of Physics, Jagiellonian University,
Prof. S. Łojasiewicza 11, PL-30348 Kraków, Poland
**3** Faculty of Physics, University of Warsaw, Pasteura 5, PL-02093 Warsaw, Poland
**4** Max Planck Institute for Solid State Research,
Heisenbergstraße 1, D-70569 Stuttgart, Germany

* krzysztof.wohlfeld@fuw.edu.pl

## Abstract

We study the problem of a single hole in an Ising antiferromagnet and, using the magnon expansion and analytical methods, determine the expansion coefficients of its wave function in the magnon basis. In the 1D case, the hole is "weakly" confined in a potential well and the magnon coefficients decay exponentially in the absence of a string potential. This behavior is in sharp contrast to the 2D square lattice where the hole is "strongly" confined by a string potential and the magnon coefficients decay superexponentially. The latter is identified here to be a fingerprint of the strings in doped antiferromagnets that can be recognized in the numerical or cold atom simulations of the 2D doped Hubbard model. Finally, we attribute the differences between the 1D and 2D cases to the magnon-magnon interactions being crucially important in a 1D spin system.

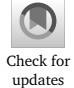

# 1  Introduction

A tendency towards particle delocalization is an ubiquitous phenomenon in quantum mechanics, for it is encoded in the Heisenberg uncertainty principle for momentum and position operators [1]. Perhaps one of its most iconic examples is the so-called particle tunneling under a finite potential barrier: even if the energy of the particle is below the potential amplitude, the probability to find a particle outside the potential well is finite. Yet, the particle is considered *localized*, for its wave function decays *exponentially* with an increasing distance from the potential well. As an important feature, the potential acts here only locally and does not increase with the distance. This example of electron localization by a potential well will be called "weak confinement" in what follows.

The fact that a particle can delocalize beyond a potential barrier has tremendous implications for the electron wave functions typically found in crystals. It allows for an electron tunnelling under the potential barrier of the periodic potential formed by the ions, leading to the modulus of the electron wave function following the periodicity of the ionic potential [2]. This is the essence of the Bloch theorem and means that an electron in a typical crystal is completely *delocalized* over all ionic sites.

Nevertheless, localization of electrons in crystals is possible. For instance, this can happen in the Mott insulators—crystals for which strong electron-electron interactions determine electron localization [3,4]. The *Mott localization* is still not fully-understood and is an area of active research—both for an integer filling [5–9], as well as in doped systems [10–20]. This lack of a complete understanding of the problem is largely due to the fact that the most widely-used models describing the problem (e.g. two-dimensional (2D) Hubbard [21], $t$–$J$ [22,23] or even the $t$–$J_z$ [4] models) cannot be solved exactly [3] and the wave function of an electron in a Mott insulator is not known in general.

Therefore, here we concentrate on perhaps the simplest, though still nontrivial and realistic,[1] problems of electron localization in the Mott insulator—the problem of the confinement of a particle by an effective potential that takes place when a single hole is added to the ordered ground state of the half-filled $t$-$J_z$ model [24–38]. Using an improved version of the recently developed magnon expansion (ME) method[2] [39–43] and analytic calculations, we unambiguously show that a single hole: (i) in 2D or higher dimensional models is "strongly" confined in the ground state and its wave-function coefficients decay *superexponentially*, i.e., much faster than in the textbook case of a single finite potential well, (ii) in a 1D chain experiences "weak" confinement, i.e., has the wave-function coefficients decaying *exponentially*, just as in a potential well. Interestingly, as we show below, these differences between the 1D and higher-dimensional cases can be easily understood in the magnon language as originating from the crucial role played by the magnon-magnon interactions in a 1D spin system. Alto-

---

[1]See the concluding section for a detailed discussion.

[2]Also known in the literature as variational approximation or momentum average [39–43].

gether, this means that lowering dimensionality and adding interactions may in fact remove "strong confinement" in favor of "weak confinement" in a strongly correlated system.

## 2  Model and methods

The Hamiltonian of the $t$–$J_z$ model reads [4]:

$$\mathcal{H} = -t \sum_{\langle ij \rangle \sigma} \left( \tilde{c}_{i\sigma}^\dagger \tilde{c}_{j\sigma} + \text{H.c.} \right) + J \sum_{\langle ij \rangle} \left( S_i^z S_j^z - \tfrac{1}{4} \tilde{n}_i \tilde{n}_j \right), \tag{1}$$

where $\tilde{n}_i = \sum_\sigma \tilde{c}_{i\sigma}^\dagger \tilde{c}_{j\sigma}$. It describes the constrained hopping of fermions $\propto t$, in the restricted Hilbert space without double occupancies, i.e., $\tilde{c}_{i\sigma}^\dagger \equiv c_{i\sigma}^\dagger (1 - n_{i\bar{\sigma}})$, along the bond $\langle ij \rangle$, and the antiferromagnetic (AF) exchange $\propto J > 0$ between the $z$-th components of the $S = \frac{1}{2}$ spins of the localized $c$-electrons [44, 45]. In what follows we study the properties of the eigenstates of the Hamiltonian (1) for the case of a single hole introduced into the half-filled limit (i.e., one electron localized at each site). As in the half-filled case, the ground state of the model (1) is an Ising antiferromagnet; the problem studied here is that of a propagation of a single hole in an Ising antiferromagnet.

The method used in the paper requires first to express the above model (1) in the magnon language. To this end, the magnons are introduced here by means of the Holstein-Primakoff transformation [46] which maps interacting spins on a boson problem,

$$S_i^z = \pm \left( \tfrac{1}{2} - n_i \right), \quad S_i^\pm = b_i, \quad S_i^\mp = b_i^\dagger. \tag{2}$$

Here $n_i = b_i^\dagger b_i$, $b_i^\dagger$ is a magnon creation operator, and the sign $\pm$ alternates between the AF sublattices in the Ising antiferromagnet. At the same time, removing a spin generates a hole: for ↑-sublattice:

$$\tilde{c}_{i\uparrow} = h_i^\dagger, \quad \tilde{c}_{i\downarrow} = h_i^\dagger S_i^+, \tag{3}$$

since removing an inverted spin also requires its realignment; complementary operations generate a hole at ↓-sublattice [24]. Crucially, the maximal number of bosons and holes has to be limited to maximally one at each site $i$ (constraint *C1*). We note that in Eqs. (2) the aforementioned constraint *C1* is implicitly imposed, allowing us to omit the "square root" multipliers on the right hand side of these definitions.

As a result, we get an *exact* representation of the $t$–$J_z$ model in the magnon language for $\alpha = 1$,

$$H = t \sum_{\langle ij \rangle} \left[ h_j^\dagger h_i \left( b_j + b_i^\dagger \right) + \text{H.c.} \right] + \frac{1}{2} J \sum_{\langle ij \rangle} \mathcal{P}_j \left( -1 + n_i + n_j - 2\alpha n_i n_j \right) \mathcal{P}_i, \tag{4}$$

where $\mathcal{P}_i = 1 - h_i^\dagger h_i$. This ensures that the terms $\propto J$ are present only on bonds without holes in the $t$–$J_z$ model (constraint *C2*). Most importantly, the last term in (4) is the only magnon-magnon interaction (at $\alpha = 1$) in this Hamiltonian—a constant term active only if two magnons (i.e., inverted spins) are present on the bond $\langle ij \rangle$. It is precisely this term which is neglected in the well-known linear spin wave (LSW) theory (at $\alpha = 0$).

Our primary method, the ME, is a novel, numerical technique of solving the polaronic models, in the Green's function formalism, through the expansion of the equations of motion [40]. It has been successfully applied to several polaronic problems [41–43], including spin, orbital, and spin-orbital polarons. The central idea is that the relevant processes are limited to the

hole's neighborhood. Thus, one can perform the expansion in real space and apply a Hilbert space cutoff based on the size and spread of the bosonic cloud surrounding the hole. This greatly reduces the Hilbert space while all the states relevant to the dynamics, as well as the *C1* and *C2* constraints, are included.

On the practical level, the method consists in multiple applications of the Dyson equation

$$\mathcal{G}(\omega) = \mathcal{G}_0(\omega) + \mathcal{G}(\omega)\mathcal{V}\mathcal{G}_0(\omega), \tag{5}$$

where $\mathcal{V}$ is the interaction of the problem, and $\mathcal{G}_0(\omega)$ is the free single-particle Green's function operator corresponding to the exactly solvable, non-interacting single particle Hamiltonian. Taking the expectation value of the (full) single-particle Green's function operator $\mathcal{G}(\omega)$ operator in a single particle state, usually the Bloch state

$$|\boldsymbol{k}\rangle \equiv h_{\boldsymbol{k}}^{\dagger}|0\rangle = \frac{1}{\sqrt{N}}\sum_j e^{i\boldsymbol{k}\cdot\boldsymbol{R}_j} h_j^{\dagger}|0\rangle, \tag{6}$$

and expanding the right-hand-side of Eq. (5) yields a single equation of motion (EOM) for the Green's function.

Since the matrix element of the $\mathcal{G}_0(\omega)$ operator is in principle known, the central problem of the expansion is evaluating the effect of the interaction $\mathcal{V}$ on the given single particle state. We assume here that this interaction is a hole-boson coupling, which either creates or destroys additional bosons in the system. For instance, acting with the kinetic coupling, as found in our problem, on the Bloch state $|\boldsymbol{k}\rangle$ leads to a hole-magnon state with momentum $\boldsymbol{k}$:

$$t\sum_{i,\delta} h_{i+\delta}^{\dagger} h_i (b_{i+\delta} + b_i^{\dagger}) h_{\boldsymbol{k}}^{\dagger}|0\rangle = \frac{t}{\sqrt{N}}\sum_{j,\delta} e^{i\boldsymbol{k}\cdot\boldsymbol{R}_j} h_{j+\delta}^{\dagger} b_j^{\dagger}|0\rangle. \tag{7}$$

Therefore, the expansion procedure introduces higher order Green's functions involving states with different hole-boson spatial configurations. These Green's functions are also unknown and are subject to the same expansion as before. Thus, the variational expansion is controlled by the maximal number of bosons created in the system—the process is continued until the EOM system closes at the desired level of expansion. Once generated, the EOM system can be solved numerically to yield the Green's function $G(\boldsymbol{k}, \omega)$, as well as all the other generalized functions appearing in the expansion, which are crucial for recreating the wavefunction and the resulting $P_n$ string length probability distributions.

We stress that the ME method is a numerically exact method of calculating the Green's function of a particular polaronic model on a particular lattice and for a given number of magnons. Thus, when applied to a hypercubic lattice and once the calculated observables cease to change with increasing number of magnons (i.e., the method "converges"), the obtained result contains all physical processes governing the propagation of a hole in the polaronic model under study; e.g. it can include the quite-often disregarded Trugman processes [26].

## 3 Results

The central object that is calculated in this paper is the probability distribution $\{P_n\}$ of observing $n$ magnons in the wave function of a hole doped into the Ising antiferromagnet. This is achieved by calculating the coefficients of the expansion of the ground state wave function of the half-filled $t$–$J_z$ model (1) with a single added hole in the above-explained "magnon language" basis [47] using the ME method in the numerically converged case, which requires keeping up to ca. 100 bosons in the calculations.

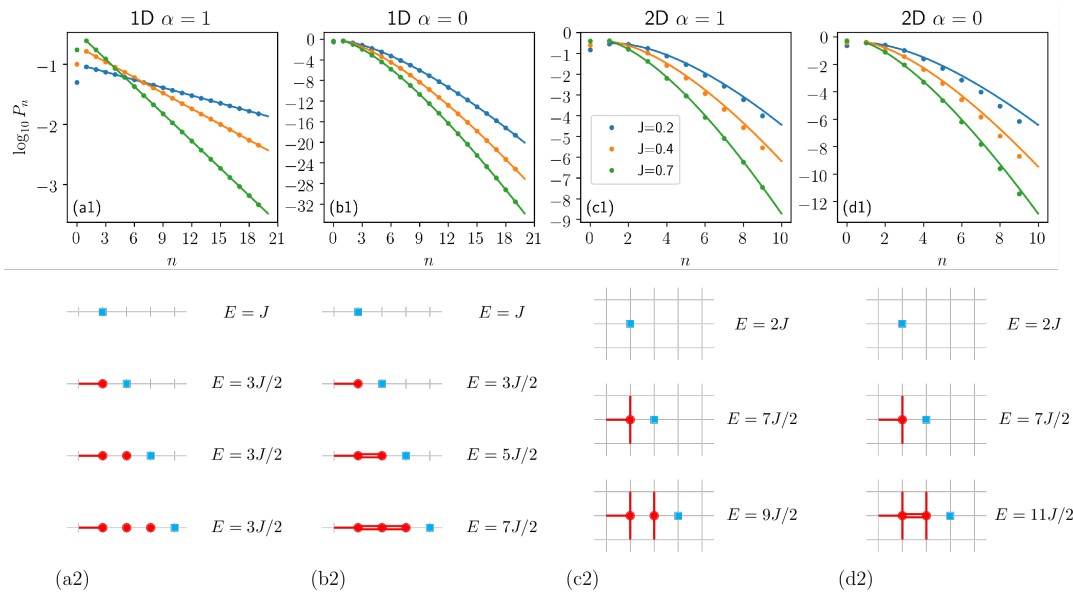

Figure 1: Probabilities $\{P_n\}$ of observing $n$ magnons in the ground state of single hole wave function as calculated using the ME method for different values of $J/t$ in the $t$–$J_z$ model in the 1D case (a1) with and (b1) without magnon-magnon interactions; and the 2D case (c1) with and (d1) without magnon-magnon interactions. In (a1) [(b1)] the curves are obtained analytically (see text); in (c1) [(d1)] the curves are fits to the numerical results (points) assuming the same functional dependence as in (b1), cf. Appendix A. The cartoons, (a2)–(d2), illustrate the differences between these four cases and show the energy cost $E$ associated with the disruption of the AF background by the moving hole (blue square) which creates magnon excitations (red circles). The energy cost of a single excited bond $E = J/2$ is represented by a single red segment.

## 3.1  1D AF Ising chain

The results, obtained for the "genuine" 1D $t$–$J_z$ model, i.e., when the magnon-magnon interactions are correctly included ($\alpha = 1$), are presented in Fig. 1(a1). We observe that, irrespectively of the value of the spin exchange $J$, the probabilities $\{P_n\}$ always decay *exponentially* with the increasing number of magnons $n$. This result is further elucidated by the analytic calculations (see Appendix A) which are in perfect agreement with the above 1D numerical results, see Fig. 1(a1). Thus, in the case of the full 1D model (1) we obtain an exponential decay of the probabilities,

$$P_{n>0}(\alpha = 1) = A\exp(-n/l), \tag{8}$$

with the decay length $l$ given by the inverse of the logarithm of the ground state energy of the AF Ising chain with a single hole ($\varepsilon_{\mathrm{GS}}$); see Appendix A for the exact expressions for the parameters $\{A, l, \varepsilon_{\mathrm{GS}}\}$. For completeness, let us note that we could have obtained this exact result also in the spinon language [36]: in this case the ground state of the AF Ising chain with a single hole is most easily understood as a bound state of a mobile holon with a spinon, which arises as a result of the single hole being confined in a 1D potential well. Nevertheless, for a better comparison with the numerics as well as with the 2D case studied below we employ here the magnon language.

Interestingly, especially when contrasted with the 2D studies below, the above exponential decay is *not* obtained when the magnon-magnon interactions are not correctly taken into account in the magnon language expression for the 1D $t$–$J_z$ model, i.e., when $\alpha \in [0, 1)$ in Eq. (4).

Instead a qualitatively "faster" decay is then clearly observed in the numerical ME calculations, see Fig. 1(b1). Again the numerical behavior is exactly reproduced by the analytically derived values of the probabilities $\{P_n\}$, see Fig. 1(b1) and Appendix A. In the quantitatively simplest case of $\alpha = 0$ the expression for $\{P_n\}$ reads:

$$P_{n>0}(\alpha = 0) = B\left[\mathcal{J}_{\frac{1}{2}-\frac{\varepsilon_{\text{GS}}}{J}+n}\left(\frac{2t}{J}\right)\right]^2 \simeq B\left|\left(\frac{t}{J}\right)^n \Gamma^{-1}\left(\frac{1}{2}-\varepsilon_{\text{GS}}+n+1\right)\right|^2, \qquad (9)$$

where $\mathcal{J}$ is the Bessel function of the first kind and the explicit equations for the constants $\{B, \varepsilon_{\text{GS}}\}$ are given in Appendix A. The expressions for $\{P_n\}$ for any value of $\alpha \in [0,1)$ are qualitatively similar but quantitatively more complex—they are explicitly given by Eqs. (26) in Appendix A. Thus, in the 1D case a first order quantum phase transition can be observed at $\alpha = 1$.

We point out that the decay given by Eq. (9) is well approximated by a $\Gamma$ function, i.e., the decay is faster than that of an exponential and can be described as a *superexponential*. The superexponential character of the decay is also visible in the asympotic behavior of $\ln P_n$ obtained for large $n$ in both of the above-mentioned cases (cf. Appendix A for details):

$$\ln P_n(\alpha = 1) \sim -n/l, \quad \ln P_n(\alpha < 1) \sim -2n\ln n. \qquad (10)$$

Hence, in the asymptotic limit the superexponential decay of $\ln P_n$ can be understood as being described by a function decreasing faster with increasing argument than a linear function with a negative coefficient.

## 3.2 2D AF Ising model on a square lattice

The central result of this paper is that the superexponential decay of probability distribution $\{P_n\}$ is found for the hole wave function of the 2D Ising antiferromagnet—not only when the magnon-magnon interactions are neglected but also for the "genuine" $t$–$J_z$ model with all the interactions correctly included. While such a superexponential decay is already visible from the plots presenting the numerical ME results in Figs. 1(c1) and 1(d1), we have further confirmed this behavior by fitting the numerical results with the following approximate expressions of the probability distributions $\{P_n\}$ that is valid for $\alpha \in [0,1]$, see Appendix A:

$$P_{n>0}(\alpha \leq 1) = C\left[\mathcal{J}_{-\frac{\varepsilon}{(2-\alpha)J}+n}\left(2t\sqrt{\bar{z}-1}/(2-\alpha)J\right)\right]^2, \qquad (11)$$

where $\{\bar{z}, \varepsilon\}$ are fitting parameters (note that the constant $C$ also depends on $\{\bar{z}, \varepsilon\}$; for more details, including the explicit expression for $C$, see Appendix A). Again one can obtain the asymptotic behavior for large $n$ of $\ln P_n$—which is the same as in Eq. (10), i.e.,

$$\ln P_n(\alpha \leq 1) \sim -2n\ln n, \qquad (12)$$

which further confirms that the 2D case is qualitatively similar to the 1D case without the magnon-magnon interactions. However, we note that a systematic error occurs in the 2D square lattice antiferromagnet when magnon-magnon interaction is neglected, see also below.

The approximate expression for the probability distribution $\{P_n\}$ in a square lattice 2D model [Eq. (11)] is motivated by the analytically exact expression obtained in the 1D case without the magnon-magnon interactions, see Eq. (9). Next, the 1D result can be extended to the case of the Bethe lattice for a given arbitrary coordination number $z$ and the respective ground state energy $\varepsilon_{\text{GS}}$. Finally, to account for the fact that the $t$–$J^z$ model is studied on the 2D square lattice and not on the Bethe lattice, we allow for some variation of the coordination number $z$ and the ground state energy $\varepsilon_{\text{GS}}$ and leave them as the fitting parameters $\bar{z}$ and $\varepsilon$, respectively.

# 4 Discussion

## 4.1 Intuitive understanding: cartoons and effective potential

Let us now try to gain some intuitive understanding of the results presented above. It turns out that this can be rather easily achieved by looking at the cartoon figures, showing the hole propagation in an Ising antiferromagnet in all four studied cases, see Figs. 1(a2)–(d2). Let us first concentrate on probably the easiest case, i.e., on the 1D model without magnon-magnon interactions ($\alpha = 0$ or the LSW case), cf. Fig. 1(a2). Here, a single hole hop to the nearest-neighbor site ($i+2$) generates a boson at site ($i+1$) and thus $n_{i+1} = 1$. Since after the previous step also $n_i = 1$, the term $J(n_{i+1} + n_i)/2$ in (4) increases the energy of this state by $J$ once magnon interactions are absent. Therefore, not only the further the hole moves the more magnons are created, but also an energy cost $\omega_0 \equiv J$ is paid after creating each magnon according to the LSW theory, while there is no mechanism to reduce energy due to magnons arising on the hole's path. The well-known linear string potential found in the 2D case [24–29, 31–35] is clearly observed here, see Fig. 1(a2). Crucially, such a phenomenon is absent once the magnon-magnon interactions are turned on in the 1D model [cf. Fig. 1(b2)]: in that case, even though the hole creates magnons at each step of its motion, there is no energy cost associated with this process as the $-\alpha J n_{i+1} n_i$ term in the $t$–$J_z$ Hamiltonian (4) cancels completely the energy cost in LSW approximation after creating all but the first magnon. This shows why the magnon-magnon interactions play such a unique role in the 1D case.

The above simple understanding changes to some extent in the 2D case. Here the magnon-magnon interactions are no longer qualitatively relevant, since, unlike in the 1D case, the energy associated with creating a magnon during hole motion cannot be canceled completely by the magnon-magnon interactions, see Fig. 1(c2)–(d2). This shows that the string-like picture [24–29, 31–35] is valid in the 2D model even when the magnon interactions are included. However, our analysis and the comparison with the ME approach confirms that the string energy evaluated for the 2D square lattice is overestimated in the SCBA and could be corrected by including the effects of magnon-magnon interactions within the modified-SCBA method [31]. *De facto*, an unphysical part of string is generated on the hole path itself, pretty much the same as in the 1D model. It is removed when the magnon-magnon interactions are included.

The above discussion can be rationalized in the language of the hole being mobile in an effective potential. Then the "genuine" 1D case (i.e., with magnon-magnon interactions correctly included) corresponds to a hole effectively moving in a potential well. On the other hand, all other cases (and in particular the 2D case with the magnon-magnon interactions) can be approximated by a hole being mobile in a (discrete version) of a linear "string-like" potential. Thus, the exponential (superexponetial) decay of the wave functions coefficients can be associated with a hole moving in a potential well (a linear potential), respectively.

Although we just referred above to the *discrete version* of the linear potential, we would like to emphasize that there exists actually a difference in the asymptotic behavior of $\{P_n\}$ in the discrete and continuous versions of the linear potential. Whereas the former was discussed above [cf. Eq. (12)] and is of the $-n \ln n$ form, the asymptotic behavior of the logarithm of the Airy function (which is the ground state wave function of a hole in the continuous linear potential [25]) is different and is given by $\sim -\frac{2}{3} n^{3/2}$. Nevertheless, in both cases the logarithms of both asymptotes "decay faster" than a linear function and are thus understood as superexponential.

## 4.2 Optical lattice experiments

The recent optical lattice experiments [48–50], as well as numerical simulations of the 2D doped Hubbard model [51, 52], reported on the histograms of the lengths of strings of mis-

aligned spins in the ground state hole wave function. As this quantity can be reliably approximated by the aforementioned probability distribution $\{P_n\}$, this means that a detailed study of the functional form of such histograms can be used to verify to what extent the hole is confined in the 2D Hubbard model.

More precisely, this work allows us to formulate a condition for the observation of a linear string potential acting on the hole in the 2D doped Hubbard (or $t$–$J$) models. First, we note that the case with the presence (absence) of the linear string potential is actually *naturally* defined in the 1D $t$–$J_z$ model with (without) interactions, respectively.[3] Second, as discussed in detail above, once the linear string potential is present (absent) in the latter model, the coefficients of the ground state wave function for a hole in the "magnon language" basis (i.e., $\{P_n\}$) need to decay superexponentially (exponentially), respectively. Therefore, combining these two observations we can conclude that, if the above-mentioned histograms observed in the optical lattice experiments: (i) showed a superexponential decay with the growing string length $l$, then this would strongly indicate that a linear string potential indeed plays a dominant role in the hole motion in the 2D doped Hubbard model; (ii) showed merely an exponential decay, then the presence of a linear string potential would be ruled out.

Can we apply the above condition to verify the existence of the linear string potential in the recent experimental or large-scale numerical simulations of the Hubbard model? While the latest optical lattice experiments [48–50] still show too few data points to unambiguously conclude whether the observed dependence is exponential or superexponential, we suggest that future optical lattice experiments might be capable of delivering more conclusive data. Moreover, the probability distribution $\{P_n\}$ can in principle be easily calculated in the (future) large-scale numerical simulations of the doped $t$–$J$ or Hubbard models.

### 4.3 Spectral functions

The above calculations show how the ground state properties of the $t$–$J_z$ model depend on the dimensionality of the problem as well as on the inclusion of the magnon-magnon interactions. This has implications primarily for the optical lattice experiments. However, also the excited states are affected in a somewhat similar manner and this has implications for the understanding of the spectral function $A(\boldsymbol{k},\omega)$ of a number of correlated compounds.

Thus, we start by defining the spectral function $A(\boldsymbol{k},\omega)$ in the usual way,

$$A(\boldsymbol{k},\omega) = -\frac{1}{\pi}\,\Im G(\boldsymbol{k},\omega+i\delta), \tag{13}$$

where $G(\boldsymbol{k},\omega) = \langle\Phi_0|\tilde{c}^\dagger_{\boldsymbol{k}\sigma}(\omega-\mathcal{H}+E_0)^{-1}\tilde{c}_{\boldsymbol{k}\sigma}|\Phi_0\rangle$ is the momentum-dependent interacting electron Green's function (the spin index $\sigma$ can be suppressed here), and $|\Phi_0\rangle$ is the ground state of the model Eq. (1) in the half-filled limit (Ising antiferromagnet) with energy $E_0$. As the spectral function is calculated using the ME method, we first need to express it in the magnon language—the constrained electron Green's function transforms then into the single-particle hole Green's function, $G_h(\boldsymbol{k},\omega) = \langle\Phi_0|h_{\boldsymbol{k}}(\omega-H+E_0)^{-1}h^\dagger_{\boldsymbol{k}}|\Phi_0\rangle$.

In 1D chain the spectral function $A(\omega)$ is always momentum-independent, but as the results obtained using the ME method show, its form depends crucially on the inclusion of the magnon-magnon interactions, see Fig. 2(a). If these interactions are included, which should always be done for a "genuine" representation of the $t$–$J_z$ model, it consists of a single dispersionless $\delta$-like peak at low energy, which indicates a quasiparticle-like state with infinite mass (a bound state), accompanied by an incoherent spectrum at higher energies. On the other

---

[3] As already stated, the presence of the linear string potential is also observed in the 2D $t$–$J_z$ model—though, then such a description is already approximate, see the discussion above. Hence, it is more instructive to refer to the 1D $t$–$J_z$ model in this discussion.

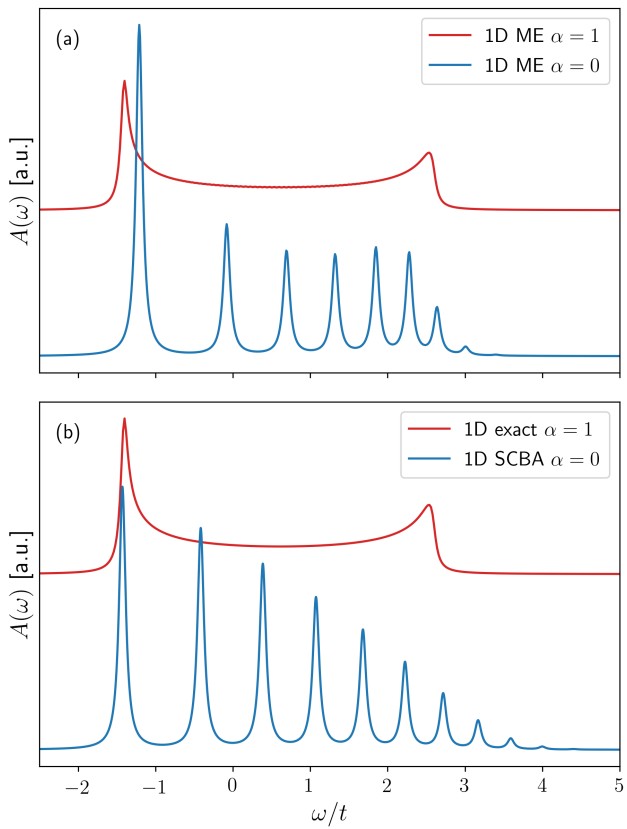

Figure 2: Spectral function $A(\omega)$ of a single hole in the 1D $t$–$J_z$ model with ($\alpha = 1$) and without ($\alpha = 0$) magnon-magnon interactions correctly included in the model calculated using: (a) the ME method, (b) the analytically exact or SCBA approach. Parameters: $J = 0.4t$, broadening $\delta = 0.05t$; the bound state at the low-energy onset of $A(\omega)$ splits off from the continuum at larger $J/t$ (lower $\delta$). Note that in the SCBA calculations for $\alpha = 0$ ($\alpha = 1$) constraints *C1*, *C2* are excluded (included), respectively.

hand, when this interaction is switched off in the LSW approximation [by putting $\alpha = 0$ in (4)], the calculated 1D spectral function is ladder-like, suggesting that the string-like potential builds up [24, 28, 29, 31], see Fig. 2(a). This striking difference between the 1D spectral function, with / without magnon-magnon interactions is also recovered using other methods:

First, let us turn to the result obtained for the "genuine" representation of the $t$–$J_z$ model with the magnon interactions correctly included. In this case, taking advantage of an exact analytical result for the $t$–$J_z$ model obtained in the spinon language and using the continued fractions [36–38], we observe that the *same* spectrum is obtained as that of the full model in the ME method, see Fig. 2(b). This shows that the exact result is indeed fully recovered using the magnon language—provided that the magnon interactions are properly taken into account. However, we emphasize that such result is only obtained in the converged case; for the non-converged ME method, i.e., once only a few or tens of bosons are retained in the ME calculations, the spectra consist of several $\delta$-like peaks (not shown).

Second, turning now to the approximate (LSW) representation of the $t$–$J_z$ model in the magnon language, we note that a ladder-like solution is also obtained when the widely-used self-consistent Born approximation (SCBA) method [24] is applied to the polaronic model with the magnon-magnon interactions switched off, see Fig. 2(b). Interestingly, this spectrum differs a bit with respect to the one obtained using the ME method, the reason for this being

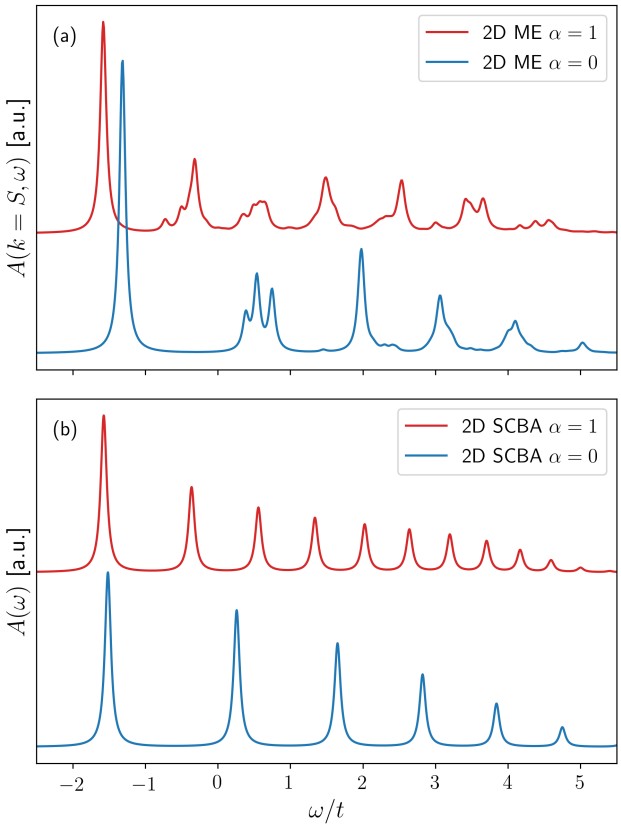

Figure 3: Spectral function $A(\boldsymbol{k}, \omega)$ (13) of a single hole in the 2D $t$–$J_z$ model with ($\alpha = 1$) and without ($\alpha = 0$) magnon-magnon interactions correctly included in the model calculated using: (a) the ME method, (b) the SCBA approach following Ref. [31]. All parameters as in Fig. 2 and $\boldsymbol{k} = S \equiv (\pi/2, \pi/2)$. Note that in the SCBA calculations for $\alpha = 0$ ($\alpha = 1$) constraints *C1*, *C2* excluded (included), respectively.

that the above-mentioned constraints *C1* and *C2* are typically implicitly neglected in the SCBA method (but are automatically taken into account in the ME method). We note that the failure of the LSW approach can be understood by comparing with the well-known 1D result [36] obtained in the "spinon language", for we have shown that one spinon corresponds here to an infinite number of magnons.

A radically different situation occurs on the 2D square lattice, as evident from the spectral functions calculated within the ME, see Fig. 3(a). Here, irrespectively of whether the magnon-magnon interactions are included or not, the spectrum is ladder-like, suggesting that the string-like potential [24–29,31–35] develops always in the 2D model: not only in the $t$–$J_z$ model approximated in terms of a polaronic model without magnon interactions (i.e., the LSW approximation) but also in the "genuine" $t$–$J_z$ model, i.e., when magnon-magnon interactions are correctly included. We note that the distances between neighboring maxima found in the ladder spectrum are lower when magnon-magnon interactions are present, see Fig. 3(a), which indicates that the string potential grows then in a slower way with increasing distance. This is indeed confirmed by a slower superexponential decrease of the wave function coefficients $\{P_n\}$, cf. Figs. 1(c1) and 1(d1), and can also be read off from the cartoon Figs. 1(c2) and 1(d2).

For a 2D square lattice the (converged) ME method is the numerically exact method and, since an analytically exact solution does not exist in this case, it can be used as a benchmark for the other more approximate methods. In particular, the 2D problem can be approximately

solved by implementing the SCBA equations derived in Ref. [31]—this time not only without the magnon-magnon interactions and *C1*, *C2* constraints excluded but also with the magnon-magnon interactions and the *C1* and *C2* constraints included, cf. Fig. 3(b). While the SCBA results qualitatively agree with the ME spectra (e.g. the SCBA also has lower distances between neighboring maxima in the ladder spectrum when magnon-magnon interactions are present), there exist two differences between these two results: (i) the higher energy peaks contain incoherent spectral weight in the ME method whereas they are of delta–like ("quasiparticle") character on the SCBA level; (ii) although the energy of the ground state in the ME method and in the SCBA method (for $\alpha = 1$ and the "canonical" value of $J = 0.4t$) is basically the same at $k = (\pi/2, \pi/2)$ point ($E = -1.58t$), there is a small difference between the two results at, e.g., $k = (0, 0)$ point ($\delta E = 0.05t$, since according to the ME method the ground state energy reads then $E = -1.63t$; unshown), in agreement with Ref. [31] which suggests a slight variance between the SCBA and numerical methods once $k \neq (\pi/2, \pi/2)$ and $J = 0.4t$. We attribute these differences to the important role played by the closed loops (Trugman loops [26]) in the 2D square lattice.[4] The latter, which are neglected by definition within the SCBA, lead to the hole propagating by "cutting the strings" and thus "disrupting" the string potential—which partially destroys the observed ladder-like spectrum by adding a small incoherent weight to the higher energy peaks.

The above results rather naturally lead to the following two questions:

First, a relatively important role played by the Trugman processes in obtaining the observed spectrum in 2D means that the the spectral function is momentum-dependent and the hole can be thought as delocalised [26]. Is it then justified to think of the hole in the ground state as being "confined" (as discussed above)? This paradox can be resolved by realising that the Trugman processes almost do not affect the probability distribution $\{P_n\}$ which is of the super-exponential type (see discussion in Sec. 3.2) and thus the hole in the ground state can indeed be well-described as being confined in a (discrete) linear potential.

Second, one might ask if the physics related to the spectral functions studied above is observable. There exists a number of compounds which can possibly be modeled by the $t$–$J_z$ Hamiltonian—these are predominantly the 1D Ising-like antiferromagnetic chains of $BaCo_2V_2O_8$ [53–55] and $SrCo_2V_2O_8$ [56, 57], the 2D ferromagnets with alternating orbital order found in $K_2CuF_4$ [42, 43] and $Cs_2AgF_4$ [44, 45], and maybe even *partially* the high-$T_c$ cuprates [58]. Thus, we expect that the angle resolved photoemission spectra (ARPES) obtained on these crystals should be qualitatively similar to the spectra shown in Figs. 2–3 (once the magnon-magnon interactions are included). This, however, could have already been predicted using the existing results reporting the spectral functions of the 1D [36–38] and 2D [24–35] $t$–$J_z$ model. A far more interesting consequence of the present spectral function study is merely theoretical: it shows that the qualitative differences between the 1D and 2D spectral functions originate solely in the different role played by the magnon-magnon interactions in the 1D and 2D models.

## 5  Conclusions

In this paper we investigated in detail the particle localization in a Mott insulator that takes place when a single hole effectively moves in a confining potential (hole confinement). The latter appears in the Mott insulating ground state with the Ising antiferromagnetic order and turns out to be of two kinds. In the 1D Ising antiferromagnet the confining potential acts only locally and does not increase with the distance. Hence, the hole is "weakly" confined—the coefficients decay exponentially—just as in the textbook example of a particle localized in a

---

[4] Note that such vertex corrections vanish in one dimension and hence we did not need to discuss them above.

potential well. On the other hand, in the 2D (or higher-dimensional) Ising antiferromagnet, the hole is subject to a (discrete version of the) linear string potential and "strongly" confined, for its coefficients in the magnon language basis decay superexponentially.

The obtained results have two important consequences:

*First*, on the pragmatic side, observation of a superexponential decay of the wave function coefficients in the magnon basis may serve as a fingerprint of the existence of a (long sought-after) linear string potential in the doped 2D antiferromagnets. This can be performed by a detailed analysis of the recent (and future) optical lattice [48–50] or numerical simulations [51,52] of the 2D doped Hubbard model.

*Second*, on the more abstract level, the use of the magnon language not only in 2D but also in 1D model allows us to understand the qualitative differences between the behavior of the hole in the 2D (or higher dimensions) and the 1D Ising antiferromagnets. These differences, which are not only observed in the decay of the hole wave function coefficients but also in the hole spectral functions, are attributed to the crucial role played by the magnon-magnon interactions in one dimension. In fact, in the 1D $t$-$J_z$ model with a single hole a first order quantum phase transition is observed when magnon-magnon interaction can no longer compensate the string potential felt by the mobile hole.[5]

Last but not least, this study demonstrates how a particular 1D antiferromagnetic problem can be described in the "magnon language". Such an approach can be seen as being complementary to, e.g., the application of the "spinon language" to magnons in the 2D non-frustrated antiferromagnet [59–62].

# Acknowledgments

We would like to thank Mona Berciu, Sasha Chernyshev, Kirill Povarov, and Yao Wang for very stimulating discussions. A. M. O. and K. W. thank the Stewart Blusson Quantum Matter Institute of the University of British Columbia for the kind hospitality. A. M. Oleś is also grateful for the Alexander von Humboldt Foundation Fellowship (Humboldt Research Award).

**Author contributions**  K.B. performed numerical calculations in the ME method, analyzed the data and prepared the figures; P.W. solved analytically the wave-function coefficients for the 1D antiferromagnet and performed the SCBA calculations; K.W. initiated the project; A.M.O. and K.W. discussed the results and wrote the paper.

**Funding information**  P. W., K. W., and A. M. O. acknowledge support by Narodowe Centrum Nauki (NCN, Poland) under Projects Nos. 2016/22/E/ST3/00560 and 2016/23/B/ST3/00839. K. B. acknowledges support by Narodowe Centrum Nauki (NCN, Poland) under Project No. 2015/16/T/ST3/00503.

# A   Derivation of the formulae for the probabilities $\{P_n\}$

In what follows we derive explicit formulae for the probabilities $\{P_n\}$ of observing $n$ magnons in the single hole ground state of the $t$–$J_z$ model [47] on the Bethe lattice. This result will be later applied to the case of the one-dimensional (1D) chain and to obtain an approximate formula for the $P_n$ on a two-dimensional (2D) square lattice.

---

[5]This happens for $\alpha \in [0,1)$, see Eq. (26) in Appendix A.

We start by choosing the basis, $\mathcal{B} = \{|n\rangle\}$, where $n \in \mathbb{N}$, and $|n\rangle$ denotes a normalized sum of states with $n$ magnons in a chain connecting the hole with a site at which the hole was created. In this basis the matrix of the $t$–$J_z$ Hamiltonian $\mathcal{H}$ [Eq. (1)] for both interacting and non-interacting magnons becomes tridiagonal,

$$
\mathcal{M}(\mathcal{H}) = \begin{bmatrix} a_1 & b_1 & 0 & 0 & \dots \\ b_1 & a_2 & b_2 & 0 & \dots \\ 0 & b_2 & a_3 & b_3 & \\ 0 & 0 & b_3 & a_4 & \ddots \\ \vdots & \vdots & & \ddots & \ddots \end{bmatrix}.
\tag{14}
$$

Using the Gaussian elimination procedure for a finite matrix and taking the limit $n \to \infty$, one can reduce the equation for the coefficients $v_n$ of the ground state vector $\vec{v}$ corresponding to the ground state energy $\varepsilon_{\mathrm{GS}}$,

$$
\mathcal{M}(\mathcal{H} - \varepsilon_{\mathrm{GS}}\mathcal{I})\vec{v} = 0,
\tag{15}
$$

into a recurrence relation,

$$
v_n = \begin{cases} v_0 & \text{if } n = 0, \\ -c_n v_{n-1} & \text{if } n > 0, \end{cases}
\tag{16}
$$

$$
c_n = \frac{b_n}{a_{n+1} - \varepsilon_{\mathrm{GS}} - b_{n+1}c_{n+1}}.
\tag{17}
$$

## A.1   1D with $\alpha = 1$

For the case of the matrix of the 1D $t$–$J_z$ Hamiltonian $\mathcal{H}$ with the magnon-magnon interactions included, calculated with respect to the energy $E_0$ of the half-filled system, we have $a_1 = J$, $a_2 = a_3 = \dots = \frac{3}{2}J$, $b_1 = -t\sqrt{2}$, $b_2 = b_3 = \dots = -t$, so the recurrence relation simplifies to

$$
v_n = \begin{cases} v_0 & \text{if } n = 0, \\ -c_1 v_0 & \text{if } n = 1, \\ -c_2 v_{n-1} & \text{if } n > 1, \end{cases}
\tag{18}
$$

where $c_1 = (\varepsilon_{\mathrm{GS}} - J)/(t\sqrt{2})$ and $c_2 = c_1/\sqrt{2}$. Normalization of the vector $\vec{v}$ requires $|v_0|^2$ to be equal to the residue of the Green's function at the quasiparticle energy $\varepsilon_{\mathrm{GS}} = \frac{3}{2}J - \frac{1}{2}\sqrt{J^2 + 16t^2}$ [30, 36], i.e.,

$$
|v_0|^2 = z_{-1} = \lim_{\omega \to \varepsilon_{\mathrm{GS}}} (\omega - \varepsilon_{\mathrm{GS}})G(\omega) = \frac{J}{\sqrt{J^2 + 16t^2}}.
\tag{19}
$$

Finally, we obtain analytical expressions for the probability $P_n$ of finding $n$ magnons in the single hole ground state,

$$
P_n = \begin{cases} |v_0|^2 = \frac{J}{\sqrt{J^2+16t^2}} & \text{if } n = 0, \\ 2|c_2|^{2n}|v_0|^2 = \frac{2J}{\sqrt{J^2+16t^2}}\left(\frac{J-\sqrt{J^2+16t^2}}{4t}\right)^{2n} & \text{if } n > 0. \end{cases}
\tag{20}
$$

The above geometric progression can also be expressed in terms of the exponential function,

$$
P_{n>0} = A\exp\left(-\frac{n}{l}\right),
\tag{21}
$$

where $A = 2|v_0|^2$ and $l^{-1} = 2\ln(|c_2|)$.

It is straightforward to observe that the asymptotic behavior for large $n$ is $\ln P_n \sim -\frac{n}{l}$.

## A.2  1D with $\alpha < 1$

For the case where the interactions between magnons are not correctly included ($\alpha < 1$), the coefficients of the matrix become $a_1 = J$, $a_{n>1} = \left((1-\alpha)(n-2) + \frac{3}{2}\right)J$, $b_1 = -t\sqrt{2}$, $b_2 = b_3 = \ldots = -t$. Therefore, the recurrence relation cannot be simplified but one can express the coefficients $c_n$ as continued fractions which, due to the linear dependence appearing on the diagonal, can be further expressed (cf. Ref. [29] for details) in terms of the Bessel functions of the first kind [63],

$$c_{n>1} = \frac{b_2}{a_{n+1} - \varepsilon_{\mathrm{GS}} - \frac{b_2^2}{a_{n+2} - \varepsilon_{\mathrm{GS}} - \ldots}} = -\frac{\mathcal{J}_{\frac{1-2\frac{\varepsilon_{\mathrm{GS}}}{J}}{2(1-\alpha)} + n - 1}\left(\frac{2t}{(1-\alpha)J}\right)}{\mathcal{J}_{\frac{1-2\frac{\varepsilon_{\mathrm{GS}}}{J}}{2(1-\alpha)} + n - 2}\left(\frac{2t}{(1-\alpha)J}\right)}, \tag{22}$$

$$c_1 = \frac{b_1}{a_2 - \varepsilon_{\mathrm{GS}} - b_2 c_2} = -\frac{\mathcal{J}_{\frac{3-2\frac{\varepsilon_{\mathrm{GS}}}{J}}{2(1-\alpha)}}\left(\frac{2t}{(1-\alpha)J}\right)}{\mathcal{J}_{\frac{3-2\frac{\varepsilon_{\mathrm{GS}}}{J}}{2(1-\alpha)} - 1}\left(\frac{2t}{(1-\alpha)J}\right)}\sqrt{2}. \tag{23}$$

The ground state energy $\varepsilon_{\mathrm{GS}}$ must satisfy the relation, $\varepsilon_{\mathrm{GS}} - J = \Sigma(\varepsilon_{\mathrm{GS}})$, with

$$\Sigma(\omega) = -2t\frac{\mathcal{J}_{\frac{3-2\frac{\omega}{J}}{2(1-\alpha)}}\left(\frac{2t}{(1-\alpha)J}\right)}{\mathcal{J}_{\frac{3-2\frac{\omega}{J}}{2(1-\alpha)} - 1}\left(\frac{2t}{(1-\alpha)J}\right)}. \tag{24}$$

Similarly to the interacting case the normalization of the ground state vector is given by the value of the residue of the Green's function at the pole $\omega = \varepsilon_{\mathrm{GS}}$,

$$|v_0|^2 = \lim_{\omega \to \varepsilon_{\mathrm{GS}}} \frac{1}{1 - \frac{d}{d\omega}\Sigma(\omega)}. \tag{25}$$

Here, neither $\varepsilon_{\mathrm{GS}}$ nor $|v_0|^2$ are given explicitly in the non-interacting case but given an expression for $\Sigma(\omega)$ one can calculate them numerically. Finally, using the recurrence relation for $v_n$, most of the Bessel functions cancel out which leads to a simple formula for the probability $P_n$ of finding $n$ magnons in the single hole ground state,

$$P_n = \begin{cases} |v_0|^2 & \text{if } n = 0, \\ 2|v_0|^2 \left(\dfrac{\mathcal{J}_{\frac{3-2\frac{\varepsilon_{\mathrm{GS}}}{J}}{2(1-\alpha)} + n - 1}\left(\frac{2t}{(1-\alpha)J}\right)}{\mathcal{J}_{\frac{3-2\frac{\varepsilon_{\mathrm{GS}}}{J}}{2(1-\alpha)} - 1}\left(\frac{2t}{(1-\alpha)J}\right)}\right)^2 & \text{if } n > 0. \end{cases} \tag{26}$$

We can also approximate the above expressions in terms of the Gamma functions. This is because for $0 < x < \sqrt{n+1}$ the Bessel functions can be expanded, leading to

$$\mathcal{J}_n(2x) \simeq \frac{x^n}{\Gamma(n+1)}, \tag{27}$$

so for large $n$ the string length probabilities can be approximated as

$$P_{n>0} \simeq B\left|\frac{1}{\Gamma\left(\frac{3-2\frac{\varepsilon_{\mathrm{GS}}}{J}}{2(1-\alpha)} + n\right)}\left(\frac{t}{(1-\alpha)J}\right)^n\right|^2, \tag{28}$$

where

$$B = \frac{2|v_0|^2}{\mathcal{J}^2_{\frac{3-2\frac{\varepsilon_{\mathrm{GS}}}{J}}{2(1-\alpha)}-1}\left(\frac{2t}{(1-\alpha)J}\right)}. \tag{29}$$

Finally, one can obtain the asymptotic behavior for large $n$ of $\ln P_n \sim -2n \ln n$. This follows either from the asymptotic behavior of the Bessel function $\ln \mathcal{J}_z(2x) \sim -z \ln z$ or of the Gamma functions $\ln \Gamma(z+1) \sim z \ln z$.

Interestingly, the asymptotic behavior for large $n$ of $P_n$ is more complex. We obtain:

$$P_n \sim \frac{B}{2\pi n}\left(\frac{\beta+n}{ex}\right)^{-2(\beta+n)}, \tag{30}$$

where

$$x = \frac{t}{(1-\alpha)J}, \quad \beta = \frac{3-2\frac{\varepsilon_{\mathrm{GS}}}{J}}{2(1-\alpha)} - 1. \tag{31}$$

### A.3 2D with $\alpha \le 1$

Before we jump to the case of a 2D square lattice we would like to generalize the above 1D approach to an equivalent problem on the Bethe lattice with coordination number $z > 2$. Once again, using the basis $\mathcal{B}$, we end up with the tridiagonal form of the matrix of the Hamiltonian. For any $\alpha \le 1$ the coefficients of the matrix are $a_1 = \frac{zJ}{2}$, $a_{n>1} = \left((\frac{z}{2}-\alpha)(n-2)+z-\frac{1}{2}\right)J$, $b_1 = -t\sqrt{z}$, $b_2 = b_3 = \ldots = -t\sqrt{z-1}$. The equivalent expressions for $c_n$ reads,

$$c_{n>1} = \frac{b_2}{a_{n+1}-\varepsilon_{\mathrm{GS}}-\frac{b_2^2}{a_{n+2}-\varepsilon_{\mathrm{GS}}-\ldots}} = -\frac{\mathcal{J}_{\frac{2z-1-2\frac{\varepsilon_{\mathrm{GS}}}{J}}{2(\frac{z}{2}-\alpha)}+n-1}\left(\frac{2t\sqrt{z-1}}{(\frac{z}{2}-\alpha)J}\right)}{\mathcal{J}_{\frac{2z-1-2\frac{\varepsilon_{\mathrm{GS}}}{J}}{2(\frac{z}{2}-\alpha)}+n-2}\left(\frac{2t\sqrt{z-1}}{(\frac{z}{2}-\alpha)J}\right)}, \tag{32}$$

$$c_1 = \frac{b_1}{a_2-\varepsilon_{\mathrm{GS}}-b_2 c_2} = -\frac{\mathcal{J}_{\frac{2z-1-2\frac{\varepsilon_{\mathrm{GS}}}{J}}{2(\frac{z}{2}-\alpha)}}\left(\frac{2t\sqrt{z-1}}{(\frac{z}{2}-\alpha)J}\right)}{\mathcal{J}_{\frac{2z-1-2\frac{\varepsilon_{\mathrm{GS}}}{J}}{2(\frac{z}{2}-\alpha)}-1}\left(\frac{2t\sqrt{z-1}}{(\frac{z}{2}-\alpha)J}\right)}\sqrt{z}, \tag{33}$$

with $\varepsilon_{\mathrm{GS}}$ satisfying the relation, $\varepsilon_{\mathrm{GS}} - \frac{zJ}{2} = \Sigma(\varepsilon_{\mathrm{GS}})$, where

$$\Sigma(\omega) = -\frac{zt}{\sqrt{z-1}}\frac{\mathcal{J}_{\frac{2z-1-2\frac{\omega}{J}}{2(\frac{z}{2}-\alpha)}}\left(\frac{2t\sqrt{z-1}}{(\frac{z}{2}-\alpha)J}\right)}{\mathcal{J}_{\frac{2z-1-2\frac{\omega}{J}}{2(\frac{z}{2}-\alpha)}-1}\left(\frac{2t\sqrt{z-1}}{(\frac{z}{2}-\alpha)J}\right)}. \tag{34}$$

The above result, with $z = 4$ and $\alpha = 1$, is equal to the self-energy calculated using Eqs. (21-23) in Ref. [31]: one merely needs to substitute in Eqs. (21-23) $\varepsilon \to \varepsilon - 2J$. This change is due to the differently defined zero energy level: in Ref. [31] the zero energy level corresponds to the Ising antiferromagnet with one hole whereas in the present paper the zero energy level corresponds to the Ising antiferromagnet.

Similarly to the 1D case the normalization of the ground state vector is given by the value of the residue of the Green's function at the pole $\omega = \varepsilon_{\mathrm{GS}}$,

$$|v_0|^2 = \lim_{\omega \to \varepsilon_{\mathrm{GS}}} \frac{1}{1 - \frac{d}{d\omega}\Sigma(\omega)}. \tag{35}$$

Also this time, using the recurrence relation for $v_n$, most of the Bessel functions cancel out which leads to a simple formula for the probability $P_n$ of finding $n$ magnons in the single hole ground state on the Bethe lattice,

$$
P_n = \begin{cases} |v_0|^2 & \text{if } n = 0, \\[2ex] \dfrac{z}{z-1}|v_0|^2 \left( \dfrac{\mathcal{J}_{\frac{2z-1-2\frac{\varepsilon_{\text{GS}}}{J}}{2(\frac{z}{2}-\alpha)}+n-1}\left(\frac{2t\sqrt{z-1}}{(\frac{z}{2}-\alpha)J}\right)}{\mathcal{J}_{\frac{2z-1-2\frac{\varepsilon_{\text{GS}}}{J}}{2(\frac{z}{2}-\alpha)}-1}\left(\frac{2t\sqrt{z-1}}{(\frac{z}{2}-\alpha)J}\right)} \right)^2 & \text{if } n > 0. \end{cases}
\tag{36}
$$

Similarly to the 1D case with $\alpha < 1$ in the case of the Bethe lattice with $z > 2$ the asymptotic behavior of $\ln P_n$ for large $n$ is $-2n \ln n$. The asymptotics for $P_n$ is more complicated. It reads

$$
P_n \sim \frac{B}{2\pi n}\left(\frac{\beta+n}{ex}\right)^{-2(\beta+n)},
\tag{37}
$$

where

$$
B = \frac{\frac{z}{z-1}|v_0|^2}{\mathcal{J}^2_{\frac{2z-1-2\frac{\varepsilon_{\text{GS}}}{J}}{2(\frac{z}{2}-\alpha)}-1}\left(\frac{2t\sqrt{z-1}}{(\frac{z}{2}-\alpha)J}\right)}, \quad x = \frac{t\sqrt{z-1}}{(\frac{z}{2}-\alpha)J}, \quad \beta = \frac{2z-1-2\frac{\varepsilon_{\text{GS}}}{J}}{2(\frac{z}{2}-\alpha)}-1.
\tag{38}
$$

In the case of a 2D square lattice, one can try to fit similar functions to the data obtained from the ME calculations assuming the same functional dependence as the one calculated for the Bethe lattice with coordination number $z > 2$ (note that for 2D square lattice we expect that $z \approx 4$). Therefore, we postulate approximate formulae for the 2D case, with $\varepsilon$ and $\bar{z}$ as the fitting parameters. They read

$$
P_n = \begin{cases} |v_0|^2 & \text{if } n = 0, \\[2ex] \dfrac{4}{3}|v_0|^2\left(\dfrac{\mathcal{J}_{-\frac{\varepsilon}{(2-\alpha)J}+n}\left(\frac{2t\sqrt{\bar{z}-1}}{(2-\alpha)J}\right)}{\mathcal{J}_{-\frac{\varepsilon}{(2-\alpha)J}}\left(\frac{2t\sqrt{\bar{z}-1}}{(2-\alpha)J}\right)}\right)^2 & \text{if } n > 0, \end{cases}
\tag{39}
$$

where the value of $|v_0|^2$ is taken as the data point corresponding to the pure hole state with no magnons. Thus, the constant $C$ mentioned in the main text in Eq. (11) reads $C = \frac{4}{3}|v_0|^2/\mathcal{J}^2_{-\frac{\varepsilon}{(2-\alpha)J}}\left(\frac{2t\sqrt{\bar{z}-1}}{(2-\alpha)J}\right)$. We note that for the considered in Fig. 1 values of $J/t$ the range of the fitted values is: $\varepsilon \in [-3.19t, -3.60t]$ ($\varepsilon \in [-1.96t, -3.09t]$) and $\bar{z} \in [1.22z, 1.57z]$ ($\bar{z} \in [1.12z, 1.30z]$) for $\alpha = 1$ ($\alpha = 0$), respectively.

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
