# Peer review of "From "Weak" to "Strong" Hole Confinement in a Mott Insulator"

_SciPost Physics, doi:SciPost Phys. 7, 066 (2019)_

## Round 4 · Referee Report · Anonymous (Referee 1) · 2019-7-29

Report

This paper examined a single hole motion of the t-Jz model in one- and two dimensions by using the magnon expansion technique. Since the underlying problem is quite old, it is important to clearly show what is new. I realized that the finding that the distribution of magnon away from the hole in two dimensions decays superexponentially is informative. Also the fact that the decay changes to exponentially in one dimension is interesting, though it is natural to see such a behavior in the presence of distance-independent string potential. Application of the results to optical lattice as well as the prediction to real materials are also important. The paper is well-written, however, there are several points to be improved.

1) In the last sentence of the abstract, it was mentioned that "Finally, we attribute ... to the peculiarities of the magnon-magnon interactions." How are the interactions peculiar?
2) In the Introduction section, it is written that "... that a single hole: ... (ii) experiences "weak" localization, ... but also certain crucial interactions present in the system are included." I could not find such an example, where crucial interactions are included. Note that this is not for the case of one dimension but more general statement.
3) Figure caption in Fig. 1 looks inconsistent with the figures.
4) In the Conclusion section, the first-order quantum phase transition was mentioned in connection with switching on and off of magnon-magnon interaction. The parameter \alpha was taken either one or zero. However, the parameter should be changed in between. What is the critical value of \alpha? What happens if you change \alpha continuously from zero to one?

---

## Round 4 · Referee Report · Anonymous (Referee 2) · 2019-7-30

Report

I have carefully read the manuscript and reviewed key references. I believe the results presented in this work are interesting and should be published in some form. However, while the authors clearly have something to say, they have also chosen a very confusing, if not a confused way of doing so. I thus suggest major revisions to their presentations before this manuscript can move any further.

The core of the present work is devoted to one of the most studied and if not well-understood, but at least better-understood problems in the strongly-correlated systems: single hole in an antiferromagnet. It has been my understanding that the problem is essentially solved, qualitatively AND quantitatively, in 1D, and qualitatively, with some good quantitative agreement with exact numerics, in 2D. The authors of the present work (a) present a new numerical treatment of the problem (coined magnon-expansion) in the Ising limit of spins, and (b) make a potentially interesting observation about “superexponential” distribution of the hole density along the “strings” in the 2D case.

However, instead of demonstrating what quantitative and qualitative advantages and insights their method (a) presents over previous results and focusing on what is the physics of the (b) feature, the author spent a significant portion of the presentation on a largely unphysical discussion of “strings in 1D” that appear “once the magnon-magnon interactions are neglected” (sic!). This discussion is largely void of a physical context and does not correspond to any physical limit of any model. It seems as if the authors are acting as hostages of their own approach, since the described setting is so clearly an artifact of the authors’ method (or SCBA) applied blindly to the problem, which, by the way, has been exactly solved more than 20 years ago, see Ref. [36].

I must also comment that the implied parallels to the problem of localization are uncalled for. In that field, such effects as interference and dephasing are important that have nothing to do with the manuscript, which simply discusses one-particle motion in two different confining potentials. Yet this name is featured in the title, abstract, and introduction, but in the end it deserves only a qualitative “looks the same”-type paragraph in the very end of the paper, from which it is clear that such a parallel is not only not instructive, but has little to no relevance to the problems at hand. As I point out in more detail below, there is more than one reason to avoid such a parallel altogether.

Requested changes

Let me proceed with the detailed comments/suggestions.
1) For the 1D problem of one hole in an AF Ising background, Ref. [36], the exact ground-state is the bound state of a fully mobile holon (hole on the AF domain wall) with a spinon (immobile AF domain wall). Somehow, a simple statement like that is avoided in text, with the only cryptic mentioning of the bound state in the caption of Fig. 2. Yet this picture would give the reader an immediate insight into the authors’ results concerning probabilities, because it maps the problem to the particle motion in a lattice-equivalent of a 1D delta-function (attractive) potential. The solution for the wave-function in this case is a textbook one, with a simple exponential decay away from the origin, and the length of such a decay given by the binding energy.
2) Instead, the reader is presented with a long-winded discussion of a dichotomy, posed by the authors as a “no-magnon-magnon” vs “magnon-magnon interactions”, which is purely artificial and is internal to the method they advocate.
3) I strongly suggest describing this picture above, listing the corresponding binding energy, which is known analytically [36], and its explicit relation to the decay length.
4) In Fig. 2, the only meaningful comparison is the one of the exact result [36] and the present method (red curves), with the explanation of any possible differences. One may mention (and may be show) that the result is unphysical (i.e., 2D-like) if one forgets about physics of the problem.

I thus suggest rewriting the 1D part in order to focus on physics and new physical insights, not on artificial problems.

5) For the 2D problem of one hole in the Ising background, regardless of the approximation, the qualitative (and qualitatively correct) picture is that of Ref. [25], more than 50 years ago. It maps the problem of the hole motion onto the motion in the linear (confining) potential. The continuum-limit solution for the wave function is that of Airy functions, I believe.
(a) I was genuinely surprised that with all the talk about “superexponential”, the authors have failed to derive a large-r (or large-n) asymptote of their probability distribution. Is it exp(-A*n*ln(n)) as it seems from Eq.(6) and from the asymptote of the Gamma-function? Basically, what does “super” stand for in this case?
Does the result agree with what the continuum solution of [25] would predict (Airy?)?
I suggest, once derived, plotting this asymptote in Fig. 1 to compare with the numerical results. This is, in my view, would be the main new result of the present study.
(b) I find it strange that, while implied, there is no direct and clear statement in text that the discussed behavior of probabilities should be a characteristic feature of a state in a linear confining potential.
(c) The main difference of the two type of the hole confinement, in 1D and in 2D problem, is between the confinement in a delta-functional and in the linear potential, respectively. This has to be said, loud and clear.

6) The straightforward SCBA-like, or string approximation are known to give qualitatively correct results, but quantitatively not-so-satisfactory agreement with, say, numerics. This has been understood as a result of several things. First is what authors refer to as to the effects of magnon-magnon interaction. Taking them into an account has lead to a modified SCBA-like approximation, Ref. [31], with much improved agreement with the available exact numerics. This approximation still neglects closed (or Trugman) loops [vertex corrections] as well as some subtler corrections due to crossed or tangential paths.
(a) The authors’ method (magnon expansion), [which, by the way, needs a slightly more than a brief description, not a just list of references] does, presumably, include all possible paths for the hole motion.
Then, the most important direct comparison needed in Fig. 3 is, again, between that of the ME method with that of Ref. [31] (red curves). There is an energy difference between the two approaches for the lowest peak (ground state). Is it due to Trugman loops?
Their effect can be largely avoided by moving k-vector to (pi/2,pi/2) point as discussed in Ref. [31].
(b) There is more structure to the higher peaks in the ME approach. Can one clarify the physical reason(s) for that?
(c) One needs an explicit statement in the text on whether the closed loops (Trugman paths) are included in the ME approach of the paper. The reason is that they are well-known to be delocalizing, thus making ANY parallels to the localization problem meaningless and self-contradictory. Or the authors are working on the Bethe lattice without ever mentioning it.
(d) The representation of the hole Green’s function in terms of the ratio of Bessel’s functions with the variable in the index was first found in Ref. [29]. This has to be mentioned explicitly.

Other comments.
7) I find that the discussion of the relevance of the current work to the interpretation of the optical experiments needs to address the following differences of the t-Jz model with the Hubbard or t-J model.
(a) It is well-known that the fluctuations in the more realistic t-J model erase strings and generate a coherent hole band of width ~2J. Is there a physical reason to expect that the strings longer than l=1 can be reliably observed? Will the peaks in the spectral function survive because of some fractional powers of J/t controlling peak separation?
(b) In the yet more realistic Hubbard model, the dispersion (and delocalization) is also provided by the effective next-neighbor hoppings (correlated 3-site terms). Same questions, are there any arguments for the survival of the string picture?

8) Since the authors implement the hard-core constraint (C1) right away, there is
no need for roots in their Eq. (2).
9) Neel AF —> Ising AF. For most, Neel implies Heisenberg model.

---

## Round 5 · Referee Report · Anonymous (Referee 2) · 2019-10-30

Report

I, basically, accept all the changes in the resubmitted version as a very good to satisfactory response to all my prior comments. Some of the points that were not changed can be considered a matter of style and I should let the authors express themselves the way they prefer to do so.

However, there is one remaining point of concern. In Fig. 3 there is a noticeable disagreement between the red curves in (a) and (b) parts, with the reference to Ref. [31] in (b). The problem is that the results of Ref. [31] are essentially identical to the Exact Diagonalization (ED) results. The argument of the authors is that their curve in (a) includes Trugman paths, while [31]/(b) does not. However, ED does include Trugman paths and for that value of J/t (=0.4), the finite-size effects are minimal. I thus believe the authors are using the wrong expression for the (modified) SCBA self-energy.
The correct one is in Ref. [31], Eq.(15). I checked the Appendix A.3, Eq. (34) and the indices of the Bessel functions come out wrong, i.e., not equal to Ref.[31]. Since it is the results of Ref. [31] that the current method should be benchmarked against, the authors should correct their calculations, results, and the ensuing discussion.

---

## Round 5 · Author Response

Warnings issued while processing user-supplied markup:

  • Inconsistency: plain/Markdown and reStructuredText syntaxes are mixed. Markdown will be used.
    Add "#coerce:reST" or "#coerce:plain" as the first line of your text to force reStructuredText or no markup.
    You may also contact the helpdesk if the formatting is incorrect and you are unable to edit your text.

Reply to “Anonymous Report 2 on 2019-7-30 Invited Report”

We would like to thank the Referee for such an extensive report and for suggesting so many useful changes---we genuinely appreciate that effort. Our reply consists of two parts. First, we comment on the general statements / impressions written by the Referee in the first page of their report (see section “General statements” below). Second, we address the specific comments 1--9 (section “Specific statements”).

“General statements”

As already stated, we really appreciate the Referee’s detailed reading of the manuscript and their very insightful comments. We have identified three main general requests voiced by the Referee, which we shall address one by one:

(1) “the author spent a significant portion of the presentation on a largely unphysical discussion of “strings in 1D” that appear “once the magnon-magnon interactions are neglected” (sic!). This discussion is largely void of a physical context and does not correspond to any physical limit of any model. It seems as if the authors are acting as hostages of their own approach, since the described setting is so clearly an artifact of the authors’ method (or SCBA) applied blindly to the problem, which, by the way, has been exactly solved more than 20 years ago, see Ref. [36].”

REPLY: We must confess that this is the only statement by the Referee that we do not fully agree with.

Let us first state two (arguably) subjective remarks:

— Our point of view is that neglecting magnon-magnon interactions should not be regarded as an artifact of SCBA but is rather due to the linear spin wave approximation (which typically is a part of the SCBA, but it is the linear spin wave approximation which comes “first” and triggers the whole problem). Since the linear spin wave theory is a relatively widely used and established approximation, we would not say that we are “hostages of (...) [our] own approach”.

— We do not find neglecting magnon interactions as “academic (i.e., not corresponding to any physical limit)” as the Referee would suggest. We note that also the t-Jz model, and sometimes even the t-J model, could be regarded just as “academic” a model (e.g., from the DFT perspective).

However, what we find more important (than the above rather subjective remarks) is that using the magnon language and switching on and off the magnon interactions can have several advantages:

— To start with, we consider using the same “magnon” language both for the 1D as well as the 2D solution to be more elegant and more didactic, than using different approaches: the “spinon” domain wall language in 1D and the magnon models in 2D. Of course, as suggested by the Referee, one could just as well talk about different forms of the potentials in 1D (delta function) and in 2D (linear potential). However, such approach is then approximate in 2D, as on the square lattice the “Trugman loops et al.” make the problem more complex. Moreover, and maybe more crucially, our numerical method is expressed in the magnon-hole basis.

— Then, when the same magnon language is used in the 1D and 2D cases, it is natural to ask the question: what is the origin of such a different solution of the 1D and 2D problem? It turns out that this stems from the magnon interactions. While in 2D they are only quantitatively important, in 1D they qualitatively alter the spectral function / the probability distribution P_n. Personally, we find this result, even if “academic” and perhaps “to be expected”, interesting and really worth presenting.

— Finally, the 1D case without magnon interactions allows for studying a problem that is exactly solvable analytically and hence can be used both for benchmarking the ME numerical approach (perfect fit) as well as for deriving the exact expression for P_n in terms of the Bessel functions [Eq. (9)]. The experience gained in 1D can then be used to firstly derive the exact result for the Bethe lattice with the coordination number z>2 [Eq. (36)] and then postulate a quantitatively approximate, but qualitatively exact, expression for the “superexponential decay” in two dimensional square lattice [Eq. (11)], assuming the same functional dependence in 2D (with and without magnon interactions) as the exactly solvable 1D model without magnon interactions [Eq. (9)].

— As a side remark, we also believe that the discussion of the magnon interactions in 2D is quite interesting. While the Referee is of course completely right that the seminal paper by Bulaevskii, Khomskii et al. from 1968 is still qualitatively correct, we find it important to explicitly show to what extent the magnon-magnon interactions alter that picture quantitatively, as already suggested in Ref. [31].

We believe that the above arguments, while not well-presented in the former version of the paper would, at least partially, convince the Referee that such a result is interesting. In the hope that this would be the case, we have updated the manuscript accordingly in order to explain and highlight the above-mentioned points in a clear manner in the current version of the manuscript; for details please see the response to the “Specific statements” as well as the Summary of changes below.

(2) The authors should concentrate in their paper on “what quantitative and qualitative advantages and insights their method (a) presents over previous results and focusing on what is the physics of the (b) feature” where (a) stands for “present(ing) a new numerical treatment of the problem (coined magnon-expansion) in the Ising limit of spins” and (b) for “mak(ing) a potentially interesting observation about “superexponential” distribution of the hole density along the “strings” in the 2D case.”

REPLY: We completely agree with the above statements and are grateful to the Referee for suggesting which parts of the manuscript they find the most interesting. Thus, we have fully followed the Referee’s recommendation and in the current version of the manuscript we now:

— Describe the numerical method and its advantages in far more detail.

— Discuss the origin as well as the implications of the “superexponential” distribution in a far more extensive way.

For details please see the response to the “Specific statements” as well as the Summary of changes below.

(3) “(...) the implied parallels to the problem of localization are uncalled for (...)”

REPLY: We also agree that using the word “localization” might be misleading [due to the association with the many-body localization phenomenon; however, note that the term “localization” is used in, e.g., Phys. Rev. B 57, 6444 (1998)] and thank the Referee for this important remark. Therefore, following the Referee’s recommendation, we have now avoided using this word as much as possible. Instead, whenever appropriate, we now have replaced the term “electron localization” (and various versions of this term) with the term “hole confinement”. Indeed, the latter describes the observed phenomenon far better. We have also updated accordingly the title, the abstract, as well as the introduction and conclusion sections. In particular, we have added this crucial distinction between the words “confinement” and “localization” by writing in the introductory paragraph:

“Therefore, here we concentrate on perhaps the simplest, though still nontrivial and realistic,\footnote{See the concluding section for a detailed discussion.} problems of electron localisation in the Mott insulator---the problem of the confinement of a particle by an effective potential that takes place when a single hole is added to the ordered ground state of the half-filled $t$-$J_z$ model~[24-38].”

We also followed the Referee's suggestion and decided to remove the last paragraph of the concluding section, to avoid misunderstanding as indeed it might have sounded misleading. For details please see the Summary of changes below.

“Specific statements”

“1) For the 1D problem of one hole in an AF Ising background, Ref. [36], the exact ground-state is the bound state of a fully mobile holon (hole on the AF domain wall) with a spinon (immobile AF domain wall). Somehow, a simple statement like that is avoided in text, with the only cryptic mentioning of the bound state in the caption of Fig. 2. Yet this picture would give the reader an immediate insight into the authors’ results concerning probabilities, because it maps the problem to the particle motion in a lattice-equivalent of a 1D delta-function (attractive) potential. The solution for the wave-function in this case is a textbook one, with a simple exponential decay away from the origin, and the length of such a decay given by the binding energy. “

REPLY: Naturally, we agree that the 1D case can be easily solved using the “spinon language”. Thus, stimulated by the Referee's comment we have heavily revised the discussion of the 1D case by explicitly mentioning that this case can be easily understood in terms of a bound state of a hole and a spinon (see Summary of changes).

“2) Instead, the reader is presented with a long-winded discussion of a dichotomy, posed by the authors as a “no-magnon-magnon” vs “magnon-magnon interactions”, which is purely artificial and is internal to the method they advocate.”

REPLY: In the reply to the “general statements” we have explained in some detail why we believe that the use of the magnon language is also useful in 1D case. Nevertheless, we completely agree with the Referee that the previous version of the paper might have left a wrong impression that the case without magnon interaction is “physical” and should be taken on equal footing with the fully interacting case. Thus, in the current version of the paper, we now stress that “Interestingly, especially when contrasted with the 2D studies below, the above exponential decay is \emph{not} obtained when the magnon-magnon interactions are not correctly taken into account in the magnon language expression for the 1D $t$--$J_z$ model (…)”.

Moreover, when discussing the 2D case, we now stress that “The approximate expression for the probability distribution ${P_n}$ in a square lattice 2D model [Eq. (11)] is motivated by the analytically exact expression obtained in the 1D case without the magnon-magnon interactions, see Eq. (9).” We have also performed several other changes in the “Results” section which hopefully makes our case more transparent to the Reader (see Summary of changes).

“3) I strongly suggest describing this picture above, listing the corresponding binding energy, which is known analytically [36], and its explicit relation to the decay length.”

REPLY: Besides adding the spinon discussion (see above), we have now mentioned in the text that “the decay length $l$ [is] given by the inverse of the logarithm of the ground state energy of the Ising AF chain with a single hole ($\varepsilon_{\rm GS}$)---see Appendix~\ref{sec:appendix} for the exact expressions of ${A, l, \varepsilon_{\rm GS} }$.” (cf. Summary of changes)

“4) In Fig. 2, the only meaningful comparison is the one of the exact result [36] and the present method (red curves), with the explanation of any possible differences. One may mention (and may be show) that the result is unphysical (i.e., 2D-like) if one forgets about physics of the problem. “

REPLY: First, we would like to emphasize that there exist no differences (beyond the numerical errors) between “the exact result [36] and the present method (red curves)” — as discussed in Sec. 4.3.: “ taking advantage of an exact analytical result for the $t$--$J_z$ model obtained in the spinon language and using the continued fractions [36-38], we observe that the \textit{same} spectrum is obtained as that of the full model in the ME method, see Fig. 2(b)”.

We agree with the Referee that the “ladder spectrum” is unphysical in one dimension and one should be clear about that — we have revised the appropriate parts of Sec. 4.3. (see Summary of changes). Nevertheless, we still think that showing this “unphysical result” is of interest to the Reader. Inter alia, it shows the role played by the constraints C1 and C2 (not that important, cf. the \alpha=0 1D results obtained using ME and SCBA) and that they are far less important than the Trugman processes in 2D (cf. \alpha=0 2D results obtained using ME and SCBA) — we have added this discussion to the current version of the paper (see Summary of changes).

“I thus suggest rewriting the 1D part in order to focus on physics and new physical insights, not on artificial problems.”

REPLY: Indeed following the Referee's comment, we have heavily rewritten the 1D part, cf. Summary of changes.

“5) For the 2D problem of one hole in the Ising background, regardless of the approximation, the qualitative (and qualitatively correct) picture is that of Ref. [25], more than 50 years ago. It maps the problem of the hole motion onto the motion in the linear (confining) potential. The continuum-limit solution for the wave function is that of Airy functions, I believe.”

REPLY: We fully agree with the above comment and believe that this point of view is also reflected in the paper.

5)“(a) I was genuinely surprised that with all the talk about “superexponential”, the authors have failed to derive a large-r (or large-n) asymptote of their probability distribution. Is it exp(-Anln(n)) as it seems from Eq. (8) and from the asymptote of the Gamma-function? Basically, what does “super” stand for in this case? Does the result agree with what the continuum solution of [25] would predict (Airy?)? I suggest, once derived, plotting this asymptote in Fig. 1 to compare with the numerical results. This is, in my view, would be the main new result of the present study.”

REPLY: These are very important points and we are grateful to the Referee for raising them. Let us answer them one by one:

— Large-n asymptote: The answer here is a bit subtle, for the large-n asymptote of the Gamma function is “almost” like exp(-Anln(n)), i.e.: (i) the large-n asymptote of ln[P(n)] ~ -2n ln(n) and is given by Eq. (10) [for 1D and (12) for 2D] in the current version of the paper, (ii) however, (i) does not imply that P(n) ~ exp[-2n ln (n)]. (iii) the proper large-n asymptote of P(n) is given in Appendix [Eq. (30) or (37)] of the current version of the paper.

— “Super” stands for a decay which is faster than an exponential, i.e., that ln P(n) decreases with n faster than -n. In “our” case this can be observed either by looking at the asymptotic behavior or by approximating P_n by a Gamma function---the latter one (which is a continuous version of the factorial) “decays faster” than an exponential. We now add this discussion to the current version of the paper (see Summary of changes).

— Agreement with “Airy”: The large-n asymptote of the logarithm of the Airy function is ~ -2/3 n^{3/2} and is different than the large-n asymptote of the P(n). Thus, the continuum solution has a different asymptote than the discrete case. Nevertheless, in both cases the logarithms of both asymptotes “decay faster” than a linear function and are thus understood as “superexponential”. This discussion is also now included in the paper (see Summary of changes—note that this discussion is added in the end of the current Sec. 4.1.).

— Plotting the asymptotes in Fig. 1: While plotting the asymptotes might indeed be a good idea, in our opinion this would make the already busy plots of Fig. 1 even less legible. This is because we already compare the numerical results against the analytical formulae for ln P(n): while in 1D the agreement is perfect, in 2D there is some small discrepancy between the two approaches.

Moreover, we believe that by comparing the top panels of Fig. 1 one can immediately observe that there is a contrast between the well-known exponential decay in the 1D (alpha=1) case (linear behavior on the ln P(n) plot) and all other cases for which one observes a faster than exponential decay, i.e. a “superexponential” one. We find the latter statement to be more important than the precise form of the asymptotic behavior of ln P(n). Nevertheless, as mentioned above, in the current version of the paper that precise form of the asymptotic behaviour is also clearly mentioned in the form of Eq. (10) or (12).

5)“(b) I find it strange that, while implied, there is no direct and clear statement in text that the discussed behavior of probabilities should be a characteristic feature of a state in a linear confining potential.”

REPLY: Yes, this is a very good remark and we have implemented it in the new version of the paper (see Summary of changes---note that this discussion is added in the end of the current Sec. 4.1.; cf. first paragraph of the Conclusion section).

5)“(c) The main difference of the two type of the hole confinement, in 1D and in 2D problem, is between the confinement in a delta-functional and in the linear potential, respectively. This has to be said, loud and clear.”

REPLY: This is also a very good remark and we have implemented it in the new version of the paper (see Summary of changes---note that this discussion is added in the end of the current Sec. 4.1.; cf. first paragraph of the Conclusion section).

“6) The straightforward SCBA-like, or string approximation are known to give qualitatively correct, but quantitatively not-so-satisfactory agreement with, say, numerics. This has been understood as a result of several things. First is what authors refer to as to the effects of magnon-magnon interaction. Taking them into an account has lead to a modified SCBA-like approximation, Ref. [31], with much improved agreement with the available exact numerics. This approximation still neglects closed (or Trugman) loops [vertex corrections] as well as some subtler corrections due to crossed or tangential paths.”

REPLY: We fully agree with the above comment and believe that this point of view is also reflected in the revised paper.

6)”(a) The authors’ method (magnon expansion), [which, by the way, needs a slightly more than a brief description, not a just list of references] does, presumably, include all possible paths for the hole motion. Then, the most important direct comparison needed in Fig. 3 is, again, between that of the ME method with that of Ref. [31] (red curves). There is an energy difference between the two approaches for the lowest peak (ground state). Is it due to Trugman loops? Their effect can be largely avoided by moving k-vector to (pi/2,pi/2) point as discussed in Ref. [31].”

REPLY: First of all let us stress that, just as in the 1D case, we believe that it is worth to look at the case without magnon-magnon interactions (see reasoning above). Nevertheless, we agree with the Referee that one should better differentiate between the “physical” (interactions included) and “approximate” (interactions neglected) cases--- which we hope we have done better in the newer version of the paper (see Summary of changes).

Coming then to the main question posed above, we answer as follows: Yes, indeed, in our opinion, the difference between the ground states is due to the onset of the closed (Trugman) loops — since this, to the best of our understanding, is the only difference between the SCBA and the ME method. We have checked that indeed such a difference is slightly smaller at ${\bf k} = (\pi/ 2, \pi / 2)$ point, though it is still extremely well visible. We included the above discussion in the current version of the paper (see Summary of changes). Note that we have also decided to show the spectral function for ${\bf k} = (\pi/ 2, \pi / 2)$ in Fig. 3(a), so that the role of the Trugman loops is suppressed as much as possible.

Following the Referee's comment mentioned in the “[]” brackets we have also expanded the Methods section by including an additional paragraph which in far more detail describes the magnon expansion method.

6)”(b) There is more structure to the higher peaks in the ME approach. Can one clarify the physical reason(s) for that?”

REPLY: A comparison between the top and bottom panels of Fig. 3 suggests that the origin of the onset of the more structure to the higher peaks in the ME approach lies in the closed (Trugman) loops. While a detailed study is needed to fully understand such behaviour (which is beyond the scope of this paper), in our opinion this can be understood as a result of the hole “cutting the strings” through the closed loops and thus “disrupting” the string potential and “destroying” the “ladder” spectrum. We added a comment on this problem in the new text of the paper (see Summary of changes).

6)”(c) One needs an explicit statement in the text on whether the closed loops (Trugman paths) are included in the ME approach of the paper. The reason is that they are well-known to be delocalizing, thus making ANY parallels to the localization problem meaningless and self-contradictory. Or the authors are working on the Bethe lattice without ever mentioning it.”

REPLY: Yes, the closed loops are included in the ME approach---this method is a numerically exact method of calculating the Green's function of the polaronic model and is applied here on a “true” hypercubic lattice (1D, 2D square). I.e., it is not applied here to a Bethe lattice. We have added an explicit statement on this problem in the current version of the paper (both in the Methods section as well as in Sec. 4.3.).

Let us also note here that, as already discussed above, indeed we have basically avoided using the word “localization”. However, strictly speaking also using the word "confined" might be questionable here. Nevertheless, we observe that on the practical level the effect of the Trugman loops on the hole “deconfinement” (i.e., hole leaving the linear potential in this case) in the ground state is negligible: the probability of finding a hole is still adequately described in the 2D case by a function which describes the “superexponential” decay and the hole confinement in a linear potential.

6)“(d) The representation of the hole Green's function in terms of the ratio of Bessel's functions with the variable in the index was first found in Ref. [29]. This has to be mentioned explicitly.”

REPLY: In the new version of the paper we refer the reader to Ref. [29] when we mention that the recurrence relation can be expressed by introducing the Bessel functions (see Summary of changes).

“7) I find that the discussion of the relevance of the current work to the interpretation of the optical experiments needs to address the following differences of the t-Jz model with the Hubbard or t-J model. (a) It is well-known that the fluctuations in the more realistic t-J model erase strings and generate a coherent hole band of width ~2J. Is there a physical reason to expect that the strings longer than l=1 can be reliably observed? Will the peaks in the spectral function survive because of some fractional powers of J/t controlling peak separation? (b) In the yet more realistic Hubbard model, the dispersion (and delocalization) is also provided by the effective next-neighbor hoppings (correlated 3-site terms). Same questions, are there any arguments for the survival of the string picture?”

REPLY: To answer the above remarks, it is important to state that our way of reasoning is actually a bit different (unfortunately this was not optimally presented in the former version of the paper---there was just a short comment in the former footnote #3):

First, this work shows that in the 2D t-Jz model (or the 1D t-Jz without magnon interactions) the “P_n” has a superexponential functional dependence. Second, in such a t-Jz model the “string picture” (i.e., the one which describes the hole moving in an AF as in a string potential) is well-defined. Combining these two observations we conclude that the “P_n” showing a superexponential decay may be regarded as a signature of the “string picture” in any model. Thus, our argument is that: if “P_n” on t-J or Hubbard models (with / without longer-range hoppings, etc.) showed a superexponential decay, then this would strongly indicate that a linear string potential indeed plays a dominant role in the hole motion in the 2D doped Hubbard (or t-J) models.

Of course, it is expected (as the Referee writes) that in the Hubbard or t-J models there will be strong deviations from the “string picture”. In fact, we completely agree with the Referee that adding the spin flip terms and / or longer range hoppings (correlated or not) would strongly alter the way the hole moves in the (doped) antiferromagnets. However, the question we ask here is: how large these deviations are (especially on the qualitative level)? We stress that it is not the purpose of this work to answer this question. We only would like to provide a criterion which can tell us whether the “string picture” can be observed in the Hubbard or t-J models. It is then up to those performing the optical lattice experiments or to the large-scale simulations of the Hubbard / t-J models to apply this criterion and check the validity of the string picture in the Hubbard / t-J cases.

In order to make the above way of reasoning clear we have rewritten the respective parts of Sec. 4.2 as well as the Conclusion section., cf. Summary of changes below.

“8) Since the authors implement the hard-core constraint (C1) right away, there is no need for roots in their Eq. (2).”

REPLY: Indeed such a change might make the presentation more clear. We have followed the Referee’s suggestion and updated the text accordingly, cf. Summary of changes below.

“9) Neel AF —> Ising AF. For most, Neel implies Heisenberg model.”

Following the Referee’s suggestion we have substituted everywhere “Neel—>Ising”, cf. Summary of changes below.

Reply to “Anonymous Report 1 on 2019-7-29 Invited Report”

We are grateful to the Referee for providing such a kind and useful report, in particular by appreciating that “the distribution of magnon away from the hole in two dimensions decays superexponentially is informative”. Below we reply to the specific points raised by the Referee:

“1) In the last sentence of the abstract, it was mentioned that "Finally, we attribute ... to the peculiarities of the magnon-magnon interactions." How are the interactions peculiar?”

REPLY: Indeed, we agree with the Referee that this sentence sounds strange and is not very informative. The current version of this sentence, in which we do not use the word “peculiar” at all, reads:

“Finally, we attribute the differences between the 1D and 2D cases to the magnon-magnon interactions being crucially important in a 1D spin system.”

“2) In the Introduction section, it is written that "... that a single hole: ... (ii) experiences "weak" localization, ... but also certain crucial interactions present in the system are included." I could not find such an example, where crucial interactions are included. Note that this is not for the case of one dimension but more general statement.”

REPLY: We thank the Referee for pointing out this logical inconsistency. The current version of the last part of the Introduction reads:

“(...) we unambiguously show that a single hole: (i)~in 2D or higher dimensional models is strongly'' confined in the ground state and its wave-function coefficients decay \emph{superexponentially}, i.e., much faster than in the textbook case of a single finite potential well, (ii)~in a 1D chain experiencesweak'' confinement, i.e., has the wave-function coefficients decaying \emph{exponentially}, just as in a potential well. Interestingly, we show below that these differences between the 1D and higher-dimensional cases can be easily understood in the magnon language, as originating from the crucial role played by the magnon-magnon interactions in a 1D spin system. Altogether, this means that lowering dimensionality and adding interactions may in fact remove strong confinement'' in favor ofweak confinement'' in a strongly correlated system.”

“3) Figure caption in Fig. 1 looks inconsistent with the figures.”

REPLY: We thank the Referee for spotting these very unfortunate and confusing typos in the figure caption. We have swapped the panels of the figure so that the caption is consistent with the figure.

“4) In the Conclusion section, the first-order quantum phase transition was mentioned in connection with switching on and off of magnon-magnon interaction. The parameter \alpha was taken either one or zero. However, the parameter should be changed in between. What is the critical value of \alpha? What happens if you change \alpha continuously from zero to one?”

REPLY: We are grateful to the Referee for this very important comment. We have verified that the critical value of $\alpha=1$, i.e., the ``strong’’ confinement happens for $\alpha\in [0,1)$, since any $\alpha<1$ cannot compensate the string potential felt by the mobile hole. Thus, we have updated the final sentence of the conclusion section and also added Eq. (26) in Appendix A which shows that the magnon coefficients decay superexponentially for $\alpha<1$.

“In fact, in the 1D $t$-$J_z$ model with a single hole a first order quantum phase transition is observed when magnon-magnon interaction can no longer compensate the string potential felt by the mobile hole \footnote{This happens for $\alpha \in [0, 1)$, see Eq. (26) in Appendix \ref{sec:appendix}}.”

---

## Round 5 · List of Changes

(0) We have changed the title to better reflect the content of the paper
(we have replaced the word “localisation” to “confinement”).

(1) The abstract and introduction were changed to reflect the general concerns raised by the Referees.

(2) Eq. (2) (the Holstein-Primakoff transformation) was changed to its linearized form, following the Referee's suggestion that the effect of the square root is imposed implicitly. A remark clarifying that point was also added in the paragraph following that equation.

(3) At the end of the Methods section we have added three paragraphs that briefly explain the formalism of the ME method, without however going into all the intricate details of computing the free Green's functions and the technical task of generating the equations of motion. We believe that this formulation is now detailed enough to give an unfamiliar reader a good idea of how the procedure is executed, without overwhelming them with unnecessary technicalities.

(4) The Results section was significantly revised and extended to accommodate the Referee's comments. Subsections were introduced to keep the results organised and easier to navigate. Eqs (11) and (12) were added to improve the discussion.

(5) The discussion of the 1D system was supplemented with a description of the spinon language picture for completeness, following the Referee's suggestion. A more detailed discussion of the (super)exponential probability decay was added to clarify the presentation.

(6) The section 3.2 was added to extend the focus on the 2D Ising system, which should constitute the main focus of the paper.

(7) Section 4 "Discussion" was extended and reorganized. A new subsection 4.1 was added to contain the more intuitive points of discussion based on "cartoons". This was mostly carved out of the former introductory statements to section 4. However, two new paragraphs were added to extend the point about the asymptotic behavior of the potentials in 1D and 2D.

(8) Subsection 4.2 was extended with an additional paragraph. The previous paragraphs were slightly altered for readability and to incorporate the Referee's comments.

(9) The discussion in section 4.3 was substantially altered and extended to accommodate the Referee's objections.

(10) The Conclusions section was altered to reflect the new key points of the article.

(11) The Appendix was extended to include more information on the derivation of the 2D case. Subsections were introduced to better organize the discussion.

(12) The order of panels in Figure 1 was changed to better reflect the logic of the current discussion.

(13) In Figs 2 and 3 labels ("a" and "b") have been added.

(14) In Fig. 3 the ME data was changed and currently shows the results for k=(pi/2, pi/2). This way the influence of the Trugman loops on the spectrum is “reduced” (as requested by the Referee).

(15) Numerous small changes in text and presentation have been introduced in order to improve the readability of the article.

---

## Round 6 · Author Response

First of all we are grateful to the Referee for reading in detail our long
response and for suggesting that he / she “accept(s) all the changes
in the resubmitted version as a very good to satisfactory response to all my prior comments”.
We are also very thankful for spotting one more mistake in our
results which concerns the energy of the ground state in the 2D SCBA results
(with the magnon-magnon interactions included). To this end, we have verified that:

(1) “Our” SCBA equations for the 2D case with magnon-magnon interactions included
[e.g. Eq. (34)] are *identical* to Eqs. (21-23) of Ref. [31]. The only difference
is due to the different zero energy level (\delta \omega =2J), see discussion
in the Summary of changes [point (2)] below.

(2) The reason why the SCBA and ME results of Fig. 3 did not match
(i.e. the ground state energies were shifted) in the previous version was
due to an accidental shift by J/2 of the SCBA result. This probably must have occurred
when “playing around” with adding / removing the C1 and C2 constraints and shifting
the zero energy level. We have now corrected this mistake and, as the Referee can observe,
the ground state energies at k=(pi/2, \pi/2) point are basically the same in the SCBA
and in the ME method (in agreement with Ref. [31]).

We thank the Referee for spotting this (rather crucial) mistake!

(3) Altogether, this means that indeed the role of the Trugman loops in obtaining the
numerically exact ground state energy seems to be rather small (in agreement with [31]).
In order to account for this fact, we have modified the text of the manuscript in few places,
see Summary of changes [point (3)] below.

We have thoroughly read the latest version of the manuscript to correct for few other
small typos and errors. We believe that the submitted version can now be published
in SciPost Physics.

We would like to thank the Referee for such a careful reading of our manuscript
and for suggesting very important changes.

Sincerely,

Krzysztof Wohlfeld

/On behalf of all Authors/

---

## Round 6 · List of Changes

(1) Following the Referee comments we have verified our results and updated Fig. 3(b)
by correcting the curve showing the SCBA \alpha = 1 results which was incorrectly shifted
by \delta \omega = J/2 in the previous version of the paper.

(2) Following the Referee comments we have updated the Appendix by adding the following
two sentences, which discuss the equality between the SCBA equations used here and in Ref. [31]:

“The above result, with $z=4$ and $\alpha=1$, is equal to the self-energy calculated using
Eqs.~(21-23) in Ref.~\cite{Che99}: one merely needs to substitute in Eqs.~(21-23)
$\varepsilon \rightarrow \varepsilon - 2J$. This change is due to the differently defined zero energy level:
in Ref.~\cite{Che99} the zero energy level corresponds to the Ising antiferromagnet with one hole
whereas in the present paper the zero energy level corresponds to the Ising antiferromagnet.“

(3) Following the Referee comments we have modified two sentences in Sec. 4.3 (as well as removed
one last sentence of that paragraph) regarding the role of the Trugman loops in obtaining the numerically
exact ground state energy. These sentences now read:

“(i) the higher energy peaks contain incoherent spectral weight in the
ME method whereas they are of delta--like (``quasiparticle'') character
on the SCBA level;
(ii) although the energy of the ground state in the ME method and
in the SCBA method (for $\alpha=1$ and the ``canonical'' value of $J=0.4t$) is basically the same
at $\vect{k}=(\pi/2,\pi/2)$ point ($E=-1.58t$),
there is a small difference between the two results at, e.g., $\vect{k}=(0,0)$ point
($\delta E= 0.05 t$, since according to the ME method the ground state energy reads then $E=-1.63t$; unshown),
in agreement with Ref.~\cite{Che99} which suggests a slight variance between the SCBA
and numerical methods once $\vect{k} \neq (\pi/2,\pi/2)$ and $J=0.4t$.”

(4) We have added one last sentence to the caption of Fig. 2 which clarifies
the inclusion of the C1 and C2 constraints:

“Note that in the SCBA calculations for $\alpha=0$ ($\alpha=1$)
constraints $C1, C2$ are excluded (included), respectively. ”

(5) We have corrected minor typos throughout the text.

---

## Editorial Decision

published